# Factors influencing medical students' decisions to remain at their alma mater for postgraduate clinical training: A cross-sectional study in Japan

Keiko Hamatsu[1], Miwa Sekine [2], Koshi Kataoka[2], Yasuhiko Kiyama[3], David Aune[2], Yuji Nishizaki [1,2]*

1 Clinical Translational Science, Juntendo University Graduate School of Medicine, Tokyo, Japan, 2 Division of Medical Education, Juntendo University Faculty of Medicine, Tokyo, Japan, 3 Hongo-Ochanomizu Campus Academic Media Center, Juntendo University, Tokyo, Japan

* ynishiza@juntendo.ac.jp

## Abstract

### Background

Despite structured postgraduate clinical training programs in Japan, a considerable number of medical graduates have recently preferred community hospitals to university hospitals for postgraduate clinical training. Understanding the factors influencing this preference is crucial for university hospitals to attract and retain top medical talent.

### Objective

To investigate how faculty and hospital characteristics influence medical graduates' decisions to apply to their alma mater for postgraduate clinical training in Japan.

### Methods

This cross-sectional study was conducted in 80 university-affiliated hospitals in Japan. Data were collected from publicly available sources, including hospital websites, the Japanese Residency Matching Program, and national databases. The variables included the proportion of medical students applying to their alma mater for postgraduate clinical training, hospital characteristics, and professors' publication records in the top five medical journals. Univariate and multiple regression analyses were performed to assess factors associated with the proportion of medical students applying to their alma mater hospitals.

### Results

The average proportion of medical students applying to their alma mater for postgraduate clinical training was 62.9%. The multiple regression analysis revealed that a higher proportion of alumni professors was significantly associated with a decreased

**Data availability statement:** All relevant data are within the paper and its Supporting Information files.

**Funding:** The author(s) received no specific funding for this work.

**Competing interests:** The authors have declared that no competing interests exist.

proportion of graduates choosing their alma mater for postgraduate clinical training (p = .001). University hospitals located in urban areas were less likely to retain graduates than those in rural areas (p = .005).

## Conclusions

Compared with community hospitals, university hospitals with greater heterogeneity in faculty members' institutional origins and those located in rural areas were more successful in retaining their graduates for postgraduate clinical training. Thus, recruiting faculty from a broader range of institutions and aligning training programs with modern medical graduate preferences may enhance university hospitals' attractiveness. Addressing these factors is essential for university hospitals in Japan to secure a well-trained physician workforce and improve healthcare outcomes.

## Introduction

In many industrialized countries, structured postgraduate clinical training programs are pivotal in equipping newly graduated physicians with essential clinical skills [1]. These programs involve rotations across various medical specialties including internal medicine, surgery, and emergency medicine to ensure broad exposure to different clinical cases and patient management strategies. This early exposure is crucial, as it lays a foundation for effective practice and decision-making in various clinical settings. The structured nature of these programs in countries, such as the United States and Australia, not only prepares residents for a wide range of medical challenges but also sets a benchmark that influences global standards in medical education [2,3].

In Japan, substantial reforms in medical education and training were introduced in 2004, when the Ministry of Health, Labour, and Welfare (MHLW) implemented a mandatory 2-year postgraduate clinical training system [4,5]. This system, administered by the Japanese Residency Matching Program (JRMP), was designed to match medical students and graduates to training hospitals using a computerized algorithm based on mutual preferences [6]. The "super-rotation" system involves a structured set of rotations among seven specialties (internal medicine, surgery, emergency medicine, pediatrics, obstetrics/gynecology, psychiatry, and community-based medicine), paralleling the approach in other developed nations but tailored to Japan's unique healthcare needs and administrative structure [7].

Despite the robust structure of these programs, there is a noticeable trend among medical graduates to prefer community hospitals over university hospitals for postgraduate clinical training. The 2022 data from the JRMP revealed a significant disparity, with 63.5% of trainees opting for community hospitals, compared with only 36.5% choosing university settings [8]. This shift raises concerns about university hospitals' ability to attract and retain top medical talent, which is essential for maintaining high standards of medical education and patient care. Key factors contributing to this trend include perceived disparities in training quality, hands-on experience, and the variety of patient cases at community versus university hospitals [9,10].

To address these challenges, it is critical to examine the factors that influence the preferences of medical graduates in certain training environments. Previous studies have highlighted variables such as emergency case volume, hospital location in populous urban areas, and financial incentives as significantly correlated with the popularity of training programs [5,11]. Therefore, our study aimed to extend this understanding by exploring the role of faculty characteristics in university hospitals, particularly the proportion of professors who are alumni and the presence of female professors, and their potential impact on the attractiveness of these institutions for postgraduate clinical training. We included faculty-related variables such as the proportion of alumni professors and academic productivity, based on previous studies reporting associations between hospital educational environments and residency outcomes [7,12].

By integrating these factors into our analysis, this study sought to provide insights into how Japan's university hospitals can enhance their appeal and compete effectively with community hospitals, thereby securing a well-trained physician workforce capable of addressing the country's evolving healthcare challenges. Understanding these dynamics could offer broader implications for healthcare policy and medical education reform, ensuring that university hospitals remain vibrant centers for advanced training and professional development.

## Materials and methods

### Study design

This cross-sectional study used publicly available nationwide matching data and information extracted from the websites of individual hospitals. As the study did not involve data from individual human participants, informed consent was deemed unnecessary in accordance with the Ethical Guidelines for Medical and Biological Research Involving Human Subjects.

### Data source and measurements

All data were collected from publicly available sources for the 2022 academic year in Japan, unless stated otherwise.

**Basic data.** We collected publicly available data from the websites of each university hospital as well as from MyNavi Resident [13], ResiNavi [14] Web of Science (WOS), and Diagnosis Procedure Combination (DPC) Database [15]. The WOS is a platform of literature search databases designed to support scientific research [16], and the DPC Database is a national healthcare database in Japan that collects detailed information on hospitalized patients.

**Hospital selection.** We selected 80 hospitals affiliated with medical universities in Japan, comprising 42 national, 8 public, and 30 private hospitals. These hospitals function as academic medical centers affiliated with medical schools and serve as designated sites for postgraduate clinical training in Japan. They are commonly referred to as "university hospitals" in Japan [5]. The selection was based on information obtained from the Ministry of Education, Culture, Sports, Science, and Technology (MEXT) University Hospital webpage and the 2022 interim announcement of clinical training matching results by the Physicians' Clinical Training Matching Council in September 2022. The National Defense Medical College and Jichi Medical University were excluded owing to their unique jurisdictional or institutional restrictions on the postgraduate clinical training system.

**University hospital information.** We collected hospital information such as university sector types (national, public, or private), average daily outpatient visits, average daily discharged patients, number of beds, number of clinical departments, number of affiliated hospitals (categorized as 0, 1, or 2 or more), number of female professors (categorized as 0, 1, or 2 or more), and location (urban or rural). Urban areas were defined as the 23 special wards of Tokyo and government-designated cities, which are major cities designated by the government based on population and economic factors. A list of official English names and URLs of the included university hospitals are provided in S1 File.

**Programs of postgraduate clinical training.** We collected data on the annual salaries of resident physicians, which are expressed in units of 10 000 yen. "Salary" refers to annual income calculated from the basic salary and compensation for on-call duties, excluding bonuses and housing allowances.

**Proportion of medical students applying to their alma mater for postgraduate clinical training.** Using data provided by the Physicians' Clinical Training Matching Council [17], we calculated the ratio of medical students applying to their alma mater hospitals for postgraduate clinical training in the academic year of 2022. This ratio is determined by the following formula: number of medical students applying to their alma mater's university hospitals—including affiliated and branch hospitals with their own training programs—for postgraduate clinical training ÷ recruitment quota specific to each facility. Since the number of graduates and the number of training positions do not always match at each hospital, using the recruitment quota as the denominator allows for a fair comparison of institutional popularity [5]. The National Defense Medical College and Jichi Medical University were excluded from the analysis owing to institutional constraints that prevent graduates from freely choosing their training sites.

**Alumni ratio of professors.** We ascertained the alma maters of professors from university and hospital websites. The ratio of alumni professors was calculated as follows: number of alumni professors ÷ total number of professors.

**Proportion of female professors.** The number of female and total professors at each university hospital were obtained from publicly available university websites. On the basis of these data, we calculated the proportion of female professors.

**Years of medical practice for professors.** We ascertained the graduation years of professors from university websites and departmental pages. The years of medical practice for professors were calculated by subtracting their year of graduation from 2022. This served as a proxy for their clinical career length, as physicians in Japan typically begin postgraduate training immediately after graduation; extended gaps are rare, particularly among those who attain professorships later.

**Publication in top five medical journals by professors.** To assess the academic impact of professors, we determined whether they had published original research articles or reviews in top journals between 2018 and 2022. We applied a binary "published" or "not published" criterion using the WOS database to check for publications in the following top five medical journals: *NEJM*, *The Lancet*, *BMJ*, *JAMA*, and *Annals of Internal Medicine*.

## Variables

The outcome variable was the proportion of medical students applying to their alma mater for postgraduate clinical training, calculated as the number of applicants from the same university divided by the recruitment quota for each university hospital.

We considered the following explanatory variables based on prior literature and expert consultation: percentage of alumni professors, calculated from publicly available faculty data; years of medical practice for professors, defined as the number of years since graduation from medical school (i.e., 2022 minus graduation year); number of hospital beds, extracted from official hospital webpages; number of clinical departments, counted from hospital department listings; average daily number of discharged patients, derived from the national DPC Database and scaled per 100 patients; average daily outpatient visits, obtained from the ResiNavi website and scaled per 1000 patients; presence of female professors, categorized as none, one, or two or more; publication in top five medical journals, binary indicator based on faculty publication records; number of affiliated hospitals, categorized as none, one, or two or more; urban or rural location, based on geographic classification (government-designated cities or 23 wards in Tokyo); and type of university, classified as national/public versus private; salary for residents, annual salary (in 10 000 yen), excluding bonuses and housing allowance.

## Data analysis

Continuous variables are summarized using means and standard deviations, whereas categorical variables are expressed as frequencies and percentages. Statistical analyses were conducted to examine the association between institutional factors and the proportion of medical students applying to their alma mater for postgraduate clinical training. Univariate linear

regression analyses were first performed for each explanatory variable described in the variables section. Variables with a p-value < 0.05 in univariate models or those considered theoretically relevant were included in the multiple regression model. Multicollinearity was checked before model fitting. Regression coefficients, 95% confidence intervals (CIs), and p-values are reported. A p-value < 0.05 was considered statistically significant, and all statistical analyses were performed using SAS (version 9.4; SAS Institute Inc., Cary, NC, USA).

## Results

**Basic characteristics of the institutions.** The average proportion of medical students applying to their alma mater for postgraduate clinical training was 62.89%, whereas the average proportion of professors who were alumni of the same institution was 36.64%. Among the surveyed facilities, only 39 (less than half) had at least one female professor, with an average postgraduate experience of 32.16 years for professors. Additionally, 50 facilities, accounting for more than 60%, were published in the top five medical journals. Details on the facilities' background information are presented in Table 1.

**Table 1. Basic characteristics of university hospitals (N = 80).**

| Variable | | Mean | SD |
|---|---|---|---|
| Proportion of medical students applying to their alma mater (% of quota) | | 41.48 | 22.03 |
| Percentage of alumni professors (%) | | 34.64 | 19.87 |
| Years of medical practice for professors | | 32.16 | 1.24 |
| Average daily outpatient visits (×1000 patients) | | 1.79 | 0.69 |
| Average proportion of patients discharged daily (per 100 patients) | | 16.37 | 4.37 |
| Number of beds | | 8.06 | 1.99 |
| Number of clinical departments | | 29.9 | 4.85 |
| Annual salary for residents (million yen) | | 40.58 | 7.96 |
| Number of professors | | 0.75 | 1.01 |
| Number of female professors | | 23.66 | 5.03 |
| Average proportion of female professors (%) | | 3.1(%) | – |
| Variable | Category | Count | Percentage |
| Type of university | Public | 50 | 62.5 |
| | Private | 30 | 37.5 |
| Number of affiliated hospitals | None | 44 | 55 |
| | One hospital | 16 | 20 |
| | Two or more Hospitals | 20 | 25 |
| Female professors | None | 41 | 51.25 |
| | One professor | 24 | 30 |
| | More than two professors | 15 | 18.75 |
| Location | Urban | 34 | 42.5 |
| | Rural | 46 | 57.5 |
| Publication in the top five medical journals by professors | Yes | 50 | 62.5 |
| | No | 30 | 37.5 |

Note: Proportion of medical students applying to their alma mater for postgraduate clinical training = number of medical students applying to their alma mater for postgraduate clinical training ÷ recruitment quota of the university hospital. The ratio was calculated at the facility level. Average daily outpatient visits were obtained from the ResiNavi website and are reported in units of 1000 patients per day. These values were included for reference only, as reporting formats vary across institutions. *Proportion of female professors was calculated as the average number of female professors divided by the average total number of professors across institutions. SD, standard deviation.

## Univariate analysis

In the univariate analysis, higher proportions of alumni professors were associated with lower proportions of medical students applying to their alma mater for postgraduate clinical training (p<.001). Furthermore, facilities with larger bed capacities and a higher number of inpatients tended to have lower proportions of medical students applying to their alma mater for postgraduate clinical training (p=.006 and p=.0013, respectively). Compared with rural university settings, urban university settings exhibited lower proportions of medical students applying to their alma mater for postgraduate clinical training (p<.001), and the presence of professors who had published in the top five medical journals was associated with a lower proportion of medical students applying to their alma mater for postgraduate clinical training (p=0.013) (Table 2).

## Multiple regression analysis

The results of multiple regression analysis revealed a negative correlation between the percentage of alumni professors and the proportion of medical students applying to their alma mater for postgraduate clinical training (p=.001). Additionally, geographic location, particularly urban areas, was significantly and negatively associated with medical students

**Table 2. Results of univariate and multiple regression analyses of determinants affecting the proportion of medical students applying to their alma mater hospitals.**

| | | Univariate | | | | Multiple regression | | | |
|---|---|---|---|---|---|---|---|---|---|
| | | B coefficient | 95% CI | | *p*-value | B coefficient | 95% CI | | *p*-value |
| | | | Lower limit | Upper limit | | | Lower limit | Upper limit | |
| Percentage of alumni professors (%) | | −0.64 | −0.9 | −0.4 | <.001 | −0.45 | −0.7 | −0.2 | <.001 |
| Number of beds | | −3.64 | −6.2 | −1.1 | .006 | 1.62 | −1.8 | 5.0 | .347 |
| Number of departments | | −0.75 | −1.8 | 0.3 | .177 | | | | |
| Years of medical practice for professors | | −2.00 | −6.3 | 2.3 | .356 | −1.86 | −5.4 | 1.6 | .292 |
| Number of discharged patients | | −1.93 | −3.1 | −0.8 | .001 | −1.30 | −2.7 | 0.1 | .072 |
| Salary (100 000 yen) | | 0.57 | −0.1 | 1.2 | .093 | | | | |
| Urban/rural | Rural | Reference | | | | Reference | | | |
| | Urban | −24.78 | −34.0 | −15.6 | <.001 | −14.95 | −25.1 | −4.8 | .005 |
| Public/private | Public | Reference | | | | Reference | | | |
| | Private | 0.08 | −10.9 | 11.1 | .988 | | | | |
| Number of affiliated hospitals | None | Reference | | | | Reference | | | |
| | One hospital | −14.49 | −28.1 | −0.9 | .037 | −9.96 | −22.0 | 2.1 | .104 |
| | Two or more hospitals | −1.05 | −13.6 | 11.5 | .868 | −0.73 | −14.3 | 12.8 | .914 |
| Female professors | None | Reference | | | | Reference | | | |
| | One | −1.20 | −13.4 | 11.0 | .846 | 0.88 | −9.3 | 11.0 | .863 |
| | Two or more | −8.20 | −22.6 | 6.2 | .259 | −0.25 | −12.8 | 12.3 | .968 |
| Publication in the top five medical journals by professors | None | Reference | | | | Reference | | | |
| | Yes | −13.48 | −24.1 | −2.9 | .013 | −2.06 | −11.7 | 7.5 | .670 |

Note: The multiple regression analysis included adjustments for the following variables: percentage of alumni professors, number of beds, years of medical practice for professors, number of discharged patients, urban/rural location, presence of female professors, and publication in the top five medical journals by professors. Each variable was independently assessed for its influence on the proportion of medical students applying to their alma mater hospitals for postgraduate clinical training. Confidence intervals (95% CI) are provided for each coefficient to reflect the precision of the estimates.

applying to their alma mater for postgraduate clinical training (p = .005) (Table 2). Variables included in the multiple regression model were selected on the basis of their statistical significance in univariate analyses (p < .05) or theoretical relevance to ensure appropriate adjustment for potential confounding factors.

## Discussion

This study examined how institutional characteristics influence medical students' decisions to remain at their alma mater for postgraduate clinical training. Using data from 80 university hospitals in Japan, we conducted multiple regression analysis to identify key factors associated with students' application behavior. Students were more likely to choose their alma mater when (1) the proportion of alumni professors was lower and (2) the university hospital was located in a rural area. These findings suggest that institutional culture and geographic setting may shape medical students' residency choices, even among hospitals of the same academic level.

In Japan, university hospitals have traditionally been central to postgraduate training owing to their resources and specialization. However, many residents have recently shifted their preference toward community hospitals, which offer exposure to a broader range of common diseases and generalist experience. Prior research has also shown that residents in community hospitals outperform those in university settings on general clinical knowledge tests, e.g., the General Medicine In-Training Examination [12]. These differences may be driven by the volume and variety of clinical experiences in community settings. While this shift has been well documented in Japan, similar nationwide preferences for community hospitals over university hospitals have not been widely reported in countries such as the United States or Australia. In these countries, university-affiliated teaching hospitals remain the primary sites for postgraduate clinical training owing to their structured educational frameworks and academic prestige [18]. This contrast underscores the unique institutional and cultural context shaping Japan's residency landscape.

In this context, faculty composition may serve as a proxy for institutional culture. The observed negative correlation between the proportion of alumni professors and the proportion of students applying to their alma mater suggests that medical students may be more attracted to institutions that foster diversity in academic backgrounds. Universities with fewer alumni professors may be perceived as more open, inclusive, or meritocratic, thereby appealing to students seeking broader perspectives in their postgraduate training.

These findings build upon prior work by Nishizaki et al., who compared community and university hospitals and found structural and educational differences that influenced residency decisions. Although our study focused on university hospitals, our results demonstrate that institutional characteristics, i.e., rural location and faculty composition, can create environments within university hospitals that resemble those of community hospitals. Particularly, rural university hospitals may play a more regionally integrated role and offer broader clinical experiences than urban ones, which in turn may appeal to medical students seeking hands-on, generalist training. This internal variation within university hospitals helps to explain differential success in retaining graduates.

While university hospitals have historically been the primary sites for medical training, owing to their ample resources and concentration of specialized diseases, there has been a recent trend toward residents preferring training at community hospitals. This shift was attributed to the opportunity to gain experience with a broader range of common diseases in community settings. This trend aligns with findings of previous reports showing that resident physicians working at community hospitals score higher than those working at university hospitals on the General Medicine In-Training Examination [19], a computer-based test used to assess clinical knowledge [12]. One possible explanation for this difference is that community hospitals primarily provide care to a larger number of residents, allowing for a wider range of clinical experiences. Consequently, university hospitals have experienced a considerable outflow of talent, leading to pronounced workforce shortages.

Acquisition of physicians, particularly outstanding resident physicians, has become a critical issue globally [20,21]. With many industrialized nations facing aging populations and a decline in the number of younger individuals entering

the medical profession, the demand for healthcare professionals is outpacing their supply. This trend is particularly pronounced in Japan, which is rapidly aging, and where urbanization is further exacerbating healthcare disparities in rural areas [5]. Therefore, retaining medical graduates as their alma mater is crucial for university hospitals to secure talent and address workforce shortages.

We considered the proportion of alumni professors to be a potential indicator of institutional culture. University hospitals with a high percentage of faculty members who are also alumni may be perceived as having more homogeneous academic environments. In Japan, this pattern often reflects the traditional "Ikyoku" system (an in-house faculty recruitment structure unique to Japanese medical schools), in which universities tend to recruit their own graduates into faculty positions. Such an inward-looking recruitment practice may signal a closed institutional culture, potentially limiting students' exposure to alternative perspectives or educational approaches. This could influence how medical students assess the openness of their training environment, making them more likely to seek opportunities outside their alma mater. The negative correlation between the proportion of alumni professors and the proportion of medical students applying to their alma mater hospitals suggests that universities with faculty drawn from a broader range of institutions may be more inclined to retain their graduates for postgraduate clinical training. One plausible interpretation of this finding is that these universities foster a culture characterized by open access and merit-based recruitment practices rather than being bound by entrenched academic cliques. By contrast, a high proportion of alumni professors might create a perception of institutional homogeneity or entrenched academic networks, potentially signaling a closed or in-group culture that discourages graduates seeking broader professional exposure. Consequently, medical students at these universities may be more inclined to look beyond their alma maters for training opportunities that provide a wider range of perspectives and are free from perceived academic in-groups. The proportion of alumni professors may reflect long-standing recruitment practices rooted in the Ikyoku system, where internal promotion and the hiring of institutional graduates have been traditionally favored.

Understanding the generational characteristics of medical students will provide additional insights into these trends. Today's medical students belong to the "Z generation," a cohort raised in an era marked by ubiquitous internet access and digital devices. They predominantly engage in Social Networking Services, where they have become accustomed to open and egalitarian communication. Consequently, members of the Z generation exhibit a propensity to readily embrace various perspectives and experiences, particularly those shaped by professors with different institutional backgrounds [22], making them receptive to guidance from such mentors. This dynamic not only facilitates the acquisition of novel knowledge but also motivates medical students to seek training opportunities beyond their alma mater.

The MHLW outlines the objective of postgraduate clinical training in its guidelines, emphasizing fundamental values such as professionalism as well as the competencies and capabilities that junior residents should acquire during their 2-year clinical training, focusing on a generalist perspective [23]. However, urban university hospitals often prioritize the treatment of specialized diseases over common conditions. This focus on specialization may not align well with the training goals of new graduates seeking broader clinical experience.

However, university hospitals play a distinct role in rural settings. They are vital for providing physicians and healthcare to remote areas, fulfilling healthcare needs within their networks, and offering comprehensive education. Rural university hospitals play a crucial role in providing care for common diseases and addressing healthcare disparities in remote regions. This focus aligns with the training objectives outlined by the MHLW, making rural alma maters attractive for graduates.

Furthermore, students at many rural medical schools reside within the community, learn the fundamentals of medicine as members of the community, and later receive clinical training at local hospitals that serve community patients. Although not statistically significant, there was a trend suggesting that a lower turnover rate of hospitalized patients—i.e., the rate of patient admission and discharge—may influence medical students' decisions to choose their alma mater as a place for postgraduate clinical training. This finding suggests that the choice to remain at a graduate university as a community member may be influenced by lower patient turnover rates. This approach not only addresses staff shortages in urban

university hospitals, as described earlier, but also offers a potential solution to the challenges of medical depopulation caused by societal aging and the urban concentration of populations.

Another factor contributing to the higher preference for choosing one's alma mater for postgraduate clinical training in rural areas is Japan's long-standing issue of physician maldistribution. To tackle physician maldistribution, the MHLW established the Medical Workforce Planning Guidelines on March 29, 2019 [5,24], introducing measures such as the Physician Distribution Index [5] and the implementation of regional quota scholarships and temporary allocations [5]. Regional quota scholarships require students to serve in designated regions or specialties after graduation in exchange for financial support from underserved prefectures. These underserved prefectures, often located in rural areas or regions with a mix of urban and remote areas, tend to establish a larger number of regional quota scholarships based on the distribution index. By contrast, overrepresented prefectures, mainly urban areas, have fewer regional quotas [5,25]. Thus, the effectiveness of these countermeasures in addressing regional disparities could influence medical students' decisions to remain in rural areas for training.

While comprehensive in its analysis of the factors influencing residency choice, this study has some limitations that should be considered when interpreting the findings. First, the reliance on data from specific sources, e.g., the MyNavi Match Program for medical students and publications from MEXT, may not fully represent all demographic or regional variances within Japan. This could limit the generalizability of our conclusions to other settings or populations. Future research should consider collecting data from a broader range of institutional and regional sources, within and outside Japan, to improve generalizability and enable international comparisons. Second, the study's design was observational, which restricted our ability to establish causality between the identified factors and medical students' decisions. Although statistically significant, the associations did not imply a direct causative relationship. To reduce potential confounding and better assess causal relationships, longitudinal or interventional study designs may be beneficial in future research. Third, the quantitative approach used in this study did not capture the subjective experiences or personal motivations of individual medical students, which could have provided deeper insights into their residency choices. Qualitative methods such as interviews or focus groups could be employed in future studies to address this gap. Fourth, our findings regarding the preferences of medical students in rural versus urban settings may have been influenced by inherent biases in student selection for rural medical programs. In Japan, these programs provide financial support (e.g., tuition waivers or living stipends) from local governments in exchange for a service obligation to practice within the sponsoring region after graduation. However, these programs are limited in scale, only a small number of students per prefecture participate, and not all prefectures offer such programs. Furthermore, even within these regions, the designated practice sites may vary in their rurality. Future studies should investigate how these obligations influence actual training site choices by collecting nationwide data on the placement patterns of regional quota graduates. In addition, the effect of the Z generation characteristics on residency choice was inferred from broader generational studies and may not accurately reflect the specific attitudes or behaviors of all medical students within this cohort. The assumption that digital nativity leads to certain educational preferences could benefit from a direct empirical examination in future studies. Lastly, although previous studies suggest that the quality of training and the variety of patient cases are important determinants of hospital choice, such data are not uniformly available from public sources and were therefore not included in the present analysis. Future research should address this limitation by incorporating institution-level surveys or clinical datasets.

## Conclusions

Our investigation highlights the key factors influencing medical students' decisions to remain at their alma mater for postgraduate clinical training. Universities with a higher proportion of alumni professors and those located in urban areas are less likely to retain graduates for postgraduate clinical training. Understanding these dynamics is crucial for university hospitals to attract and retain new medical graduates. University hospitals can enhance their appeal by fostering an

environment that values faculty drawn from a broader range of institutions and aligns training programs with Z generation medical students' preferences. Additionally, addressing the imbalance between specialized and generalist training, especially in urban settings, could encourage more graduates to remain at their alma mater, contributing to a more balanced distribution of physicians and improved healthcare outcomes in Japan.

## Supporting information

**S1 File. Excel spreadsheet listing university hospitals in Japan, including their names and website URLs.**
(XLSX)

**S1 Dataset. Compressed ZIP archive containing the complete anonymized dataset utilized for statistical analyses reported in this manuscript.**
(ZIP)

## Acknowledgments

The authors would like to thank Editage (www.editage.jp) for the English language review.

## Author contributions

**Conceptualization:** Keiko Hamatsu, Yuji Nishizaki.

**Data curation:** Keiko Hamatsu, Miwa Sekine, Koshi Kataoka.

**Formal analysis:** Miwa Sekine, Koshi Kataoka.

**Funding acquisition:** Yuji Nishizaki.

**Investigation:** Keiko Hamatsu, Miwa Sekine, David Aune, Yuji Nishizaki.

**Methodology:** Yuji Nishizaki.

**Project administration:** Yuji Nishizaki.

**Resources:** Yasuhiko Kiyama, Yuji Nishizaki.

**Software:** Miwa Sekine, David Aune.

**Supervision:** Yuji Nishizaki.

**Validation:** Yasuhiko Kiyama, David Aune.

**Visualization:** Koshi Kataoka.

**Writing – original draft:** Keiko Hamatsu.

**Writing – review & editing:** Miwa Sekine, Koshi Kataoka, Yasuhiko Kiyama, David Aune, Yuji Nishizaki.

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
