## [Decision Letter · Decision Letter 0]

Dear Dr. Nishizaki,

Thank you for submitting your manuscript to PLOS ONE. After careful consideration, we feel that it has merit but does not fully meet PLOS ONE’s publication criteria as it currently stands. Therefore, we invite you to submit a revised version of the manuscript that addresses the points raised during the review process.

Dear Authors thank you for submitting an interesting article, kindly go through and address all the reviewers comments.

We look forward to receiving your revised manuscript.

Kind regards,

Ramya Iyadurai

Academic Editor

PLOS ONE

2. We note that there is identifying data in the Supporting Information file <Data.xlsx>. Due to the inclusion of these potentially identifying data, we have removed this file from your file inventory. Prior to sharing human research participant data, authors should consult with an ethics committee to ensure data are shared in accordance with participant consent and all applicable local laws.

-Location data

Please remove or anonymize all personal information, ensure that the data shared are in accordance with participant consent, and re-upload a fully anonymized data set. Please note that spreadsheet columns with personal information must be removed and not hidden as all hidden columns will appear in the published file.

Additional Editor Comments:

Dear Authors

thank you for an interesting and relevant study,

Can you please complete Table 1, the columns after average out patient visits (in 1000's) has been left blank.

kindly provide this information.

in the discussion kindly remove lines 244 - 248 they are interrupting the discussion of the results.

Kindly provide a short summary of the results in the beginning of the discussion

and discuss why the presence of alumni professors could interfere with the choice of residency,

as authors you have chosen this as a important factor to analyse, it would be useful for the readers to understand

why and how you had thought of this factor and why you think it may impact residency recruitment?

Kindly address all the comments provided by the reviewers.

Reviewers' comments:

Reviewer's Responses to Questions

**Comments to the Author**

1. Is the manuscript technically sound, and do the data support the conclusions?

Reviewer #1: Yes

Reviewer #2: Partly

2. Has the statistical analysis been performed appropriately and rigorously?

Reviewer #1: No

Reviewer #2: Yes

3. Have the authors made all data underlying the findings in their manuscript fully available?

Reviewer #1: Yes

Reviewer #2: No

4. Is the manuscript presented in an intelligible fashion and written in standard English?

Reviewer #1: Yes

Reviewer #2: Yes

Reviewer #1: Thank you for the opportunity to review the article entitled, “Factors Influencing Medical Students' Decisions to Remain at Their Alma Mater for Residency Training: A Cross-sectional Study in Japan.” This study aimed to investigate how faculty and hospital characteristics influence medical graduates' decisions to apply to their alma mater for postgraduate clinical training in Japan. The authors have collected data from different datasets and websites and chosen the related variables to determine the key factors that influence medical students’ decisions.

Although the manuscript is thoughtfully written and contributes significantly to university hospitals attracting medical applicants for PGY training, I believe it could benefit from addressing the points outlined below to further enhance its clarity and impact. I hope my comments will be helpful.

Title:

1. While the title fits well with the manuscript, I suggest using consistent terminology for "postgraduate training" throughout the document.

Abstract:

1. L.35. I think “Multivariate linear regression” is a misuse term in the abstract. Due to you have only one outcome variable (the number of medical students applying to their alma mater for postgraduate training), “multivariable linear regression” or “multiple regression” is more suitable.

2. L.31-33. In the abstract, you mentioned that this study would use the variable, the number of medical students applying to their alma mater for postgraduate training. However, in the Data analysis section, you mention that you used “the likelihood of medical…. (L.171-172)” Please make it clear.

Introduction:

1. L.74-75. “the 2022 data from the JRMP……63.5% of trainees 75 opting for community hospitals, compared with only 36.5% choosing university settings.” Is this a special phenomenon being Japan or do other countries also mention this issue? For example, in the United States and Australia, which you had mentioned in the Introduction (L.60).

2. L.84-87. Why you want to add the role of faculty characteristics as independents variables. Any reference support?

Materials and Methods:

1. What is the time period or year of the data you retrieved?

2. What do you mean by “university hospital”?

3. The terms "Rate" and "Ratio" have different statistical meanings. Please clarify this distinction and use the correct term.

4. L137-140. Why is the denominator of "the rate of medical students applying to their alma mater for postgraduate clinical training" "recruitment capacity specific to each facility" instead of the number of medical students in that year (the academic year of 2022)?

5. L.150-153. Does "Duration of medical practice" refer to "length of service/seniority"? How could you ensure that physician professors started working in the hospital immediately after they graduated?

6. Given that the quality of training and patient diversity may influence medical students’ decisions (L.78-79); did you gather this data? I recommend treating these factors as confounding variables in data analysis.

7. Please clarify the term “multivariate linear regression” or “multivariable linear regression/multiple regression.”

8. L.167-168. You mentioned “retention of medical school graduates at alma mater hospitals,” did you mean retention rate or application rate? It is a little bit confusing. I would suggest to use consistent term.

9. Please clarify and re-organize the data analysis section (L.168-176). Why you only analyzed three variables in the univariate linear regression? How did you define potential confounders? What were your independent variables (factors) in the multiple regression?

Results:

1. What was the difference between “Percentage of medical students applying to their alma mater for postgraduate clinical training (%)” and “Percentage of medical students applying to their alma mater for postgraduate clinical training/enrollment capacity (%)” in Table 1?

2. The data of “Average daily outpatient visits (1000 patients)” was missing in Table 1.

3. “recruitment quota of the university hospital (L.198)” and “recruitment capacity specific to each facility (L.140)”? Please use consistent wording.

4. For the univariate analysis, salary showed no significant relationship with the application rate. Therefore, the statement, “There was also a trend of higher salaries …for postgraduate clinical training (p = 0.0929),” may not be appropriate.

5. For the footnote of Table 2, some of the adjust variables were not significant related to the outcome variables (the application rate). Please explain the adjust variables for multivariate analysis in the Result section.

6. Please clarify the outcome variable is retention rate or application rate in content and Table 2.

7. You mentioned that you would collect the average daily outpatient visits and average daily inpatient admission, but those variables did not show in the analysis (Table 2). Additionally, you have two new variables (number of departments and number of discharge patients) in Table 2, which were not mentioned in the methods section.

8. L.222. delete “Furthermore.”

Discussion:

1. L.237-240. In my understanding, all (N=81) of your included hospitals were at the same level—university hospital. You just compare their location (urban or rural). However, Nishizaki et al. [18] compared community hospitals and university hospitals, which compared different levels of hospital. Or is the community hospitals were all in rural and the university hospitals were all in urban in Nishizaki et al.’s study?

2. L.244-248. The linkage between 6-year medical training and the institution’s concerns is unclear.

3. L.258-260. It is unclear that the correlation of university recruitment culture and your finding. Please explain it more.

4. The limitation section is comprehensive. However, I would suggest providing some solutions for each limitation for future study.

Conclusion:

1. Conclusion is well fit with the study content.

References:

1. Please recheck your reference list. Some of institution names are unclear.

2. Most of the references you use in the discussion section are old documents (10 years ago) or in Japanese; please update the cited documents to new ones and use the English references to help foreign readers understand

Other:

The writing was unclear in some areas making it difficult to follow. The authors may benefit from support with editing.

Reviewer #2: The article is interesting and well-written. From an international perspective, it reveals a reverse concern compared to many nations, in that it reveals a recruitment problem in urban areas in Japan. Most health care systems are challenged to recruit physicians to rural areas. The authors might well want to discuss there results with this international lens, focusing on likely causes of the disparity. The following issues need to be addressed:

1. The statistics of the article are well-performed. Where concerns arise regarding the data supporting conclusions, it may well be just be a matter of definition. The chief concern relates to the repeated assertion that recruitment could be helped by improved diversity of the faculty. Most readers would interpret improved diversity to mean race or gender. Race is not tested, and gender does not seem even close to being a significant contributor to recruitment. Instead, diversity seems to mean the proportion of faculty that are alumni. A better approach to describing this metric would remove confusion.

2.The article includes some interesting data that should be discussed, such as the fraction of faculty that are female. At the moment this data seems to lack a denominator. expressed as a fraction, many researchers would find this useful additional information.

3. It would be useful to include a discussion of the factors that likely control the fraction of faculty that are alumni.

4. Much of the data does not seem easily available, particularly the data from websites of each university hospital, which are not specified or linked.

**Do you want your identity to be public for this peer review?** For information about this choice, including consent withdrawal, please see our Privacy Policy

Reviewer #1: No

Reviewer #2: No

---

## [Author Response · Author response to Decision Letter 1]

12 Jun 2025

Responses to the Reviewers’ Comments

Thank you for giving us the opportunity to submit a revised draft of our manuscript, titled “Factors Influencing Medical Students' Decisions to Remain at Their Alma Mater for Postgraduate Clinical Training: A Cross-sectional Study in Japan” for publication in PLOS ONE. We appreciate the time and effort that you and the reviewers dedicated to providing feedback on our manuscript, and we are grateful for the insightful comments. We have incorporated most of the suggestions made by the reviewers. These changes are highlighted within the manuscript for your convenience. Additionally, we have provided point-by-point responses to the reviewers’ comments below in blue. All page and line numbers refer to the revised manuscript file with tracked changes.

Comments from the Editorial Office:

Comment 1: Please ensure that your manuscript meets PLOS ONE's style requirements, including those for file naming. The PLOS ONE style templates can be found at

Author response: We are grateful to the Editorial Office for the critical comments and helpful suggestions that have helped us improve our paper considerably. As indicated in the responses that follow, we have considered all these comments and suggestions in the revised version of our paper (text highlighted in pink).

Comment 2: We note that there is identifying data in the Supporting Information file <Data.xlsx>. Due to the inclusion of these potentially identifying data, we have removed this file from your file inventory.

Please remove or anonymize all personal information, ensure that the data shared are in accordance with participant consent, and re-upload a fully anonymized data set.

Author Response:

Thank you for your comment regarding data anonymization. We confirm that the dataset used in this study was compiled exclusively from publicly available sources, including university hospital websites, national databases, and official matching program results. However, we acknowledge that even publicly available information, such as program ID numbers, could potentially be used in combination with other variables to reidentify specific institutions. To fully comply with the data policy of PLOS ONE and safeguard against any possibility of indirect identification, we have removed the program ID column from the revised dataset. The updated Supporting Information file (Anonymized_Data.xlsx) has been re-uploaded and includes no direct or indirect identifiers.

Comment 3: Please review your reference list to ensure that it is complete and correct. If you have cited papers that have been retracted, please include the rationale for doing so in the manuscript text, or remove these references and replace them with relevant current references. Any changes to the reference list should be mentioned in the rebuttal letter that accompanies your revised manuscript. If you need to cite a retracted article, indicate the article’s retracted status in the References list and also include a citation and full reference for the retraction notice.

Author Response:

We have thoroughly reviewed our reference list to ensure its completeness and accuracy. No retracted papers were identified among the cited references. Additionally, we have updated and standardized the reference formatting in accordance with the style guidelines of PLOS ONE. These changes have been incorporated into the revised manuscript (pages 20–23, lines 441–511).

Comment 4: Can you please complete Table 1, the columns after average out patient visits (in 1000's) has been left blank.

kindly provide this information.

Author Response:

Thank you for your comment regarding Table 1. In the process of revising the manuscript for improved accuracy, we have updated our data source to the DPC dataset, which is a nationwide database focused on inpatient care in Japan. Because this dataset does not include outpatient statistics and its structure differs from other publicly available sources, we initially excluded outpatient visit data from the table. However, in response to your comment, we have added the average daily outpatient visits as a supplementary reference, using data obtained from the national residency information website “ResiNavi.” These values are presented in units of 1000 patients per day. A corresponding footnote has been added to Table 1 to clarify the data source and contextualize its interpretation.

Specific changes:

Table 1 (pages 10–11): We added the following row under continuous variables:

Average daily outpatient visits (�1000 patients): Mean = 1.79, SD = 0.69

Footnote to Table 1 (page 11, line 228–230):

“Average daily outpatient visits were obtained from the ResiNavi website and are reported in units of 1000 patients per day. These values were included for reference only, as reporting for-mats vary across institutions.”

Comment 5: in the discussion kindly remove lines 244 - 248 they are interrupting the discussion of the results.

Author Response:

Thank you for your suggestion. We agree that the text on those lines interrupt the flow of the Discussion. Accordingly, we have removed this portion from the revised manuscript to improve clarity and coherence in presenting our findings.

Comment 6: Kindly provide a short summary of the results in the beginning of the discussion

Author Response:

Thank you for your helpful suggestion. To improve the clarity and flow of the Discussion section, we have revised the opening paragraph to include a concise summary of the main results. This provides a clearer starting point for interpreting our findings in the broader context of previous research.

Specific change:

Page 13, lines 264–271

“This study examined how institutional characteristics influence medical students’ decisions to remain at their alma mater for postgraduate clinical training. Using data from 80 university hospitals in Japan, we conducted multiple linear regression analysis to identify key fac-tors associated with students’ application behavior. Students were more likely to choose their alma mater when (1) the proportion of alumni professors was lower and (2) the university hospital was located in a rural area. These findings suggest that institutional culture and geographic setting may shape medical students’ residency choices, even among hospitals of the same aca-demic level.”

Comment 7: and discuss why the presence of alumni professors could interfere with the choice of residency, as authors you have chosen this as a important factor to analyse, it would be useful for the readers to understand

why and how you had thought of this factor and why you think it may impact residency recruitment?

Author Response:

Thank you for this valuable comment. We agree that the rationale for including the proportion of alumni professors as a key factor should be more clearly articulated. In Japan, some university hospitals are perceived to have entrenched academic networks dominated by faculty who are alumni of the institution, and this has raised concerns that institutional cultures may become insular, potentially limiting exposure to diverse ideas and perspectives. We hypothesized that such academic homogeneity could influence medical students’ perceptions of the learning environment and mentorship opportunities, thereby affecting their decision to remain at their alma mater for postgraduate training. To clarify this point, we have revised the Discussion section to briefly explain our reasoning behind including this variable.

Specific changes:

Pages 15–16, lines 318–327

“We considered the proportion of alumni professors to be a potential indicator of institutional culture. University hospitals with a high percentage of faculty members who are also alumni may be perceived as having more homogeneous academic environments. In Japan, this pattern often reflects the traditional “Ikyoku” system (an in-house faculty recruitment structure unique to Japanese medical schools), in which universities tend to recruit their own graduates into faculty positions. Such an inward-looking recruitment practice may signal a closed institutional culture, potentially limiting students’ exposure to alternative perspectives or educational approaches. This could influence how medical students assess the openness of their training environment, making them more likely to seek opportunities outside their alma mater.”

Comment 8: Kindly address all the comments provided by the reviewers.

Author Response:

We have addressed all the comments provided by the reviewers.

Although this point was not raised by the reviewers, we recognized that the text in the Hospital selection (page 6, lines 120–130) and Proportion of medical students applying to their alma mater for postgraduate clinical training sections (page 7, lines 147–159) might lead to confusion regarding the inclusion and exclusion of certain institutions. Thus, to improve clarity and consistency for international readers, we have added brief explanations in both sections to state that the “National Defense Medical College and Jichi Medical University were excluded owing to institutional constraints that prevent graduates from participating in the general residency matching system.” This addition was made solely to enhance transparency and does not affect the study design or results.

Revised text (Page 6, lines 128–130), (page 7, lines 157–159):

The National Defense Medical College and Jichi Medical University were excluded owing to institutional constraints that prevent graduates from participating in the general residency matching system.

Further, to enhance clarity and maintain consistency across the manuscript, we have revised several variable labels in Table 1 (page 10-11). These edits ensure alignment with the definitions provided in the Variables and Data Analysis sections. Specifically, we have standardized terminology (e.g., “Years of medical practice for professors”) and clarified measurement units (e.g., “Average daily outpatient visits (per 1,000 patients)”) to avoid ambiguity. These changes were purely editorial and did not affect any statistical results or interpretations.

Revised text (Table 1, Page 10–11):

Years of medical practice for professors

Average daily outpatient visits (per 1,000 patients)

Number of clinical departments

Annual salary for residents (million yen)

Reviewer #1:

Comment 1: Thank you for the opportunity to review the article entitled, “Factors Influencing Medical Students' Decisions to Remain at Their Alma Mater for Residency Training: A Cross-sectional Study in Japan.” This study aimed to investigate how faculty and hospital characteristics influence medical graduates' decisions to apply to their alma mater for postgraduate clinical training in Japan. The authors have collected data from different datasets and websites and chosen the related variables to determine the key factors that influence medical students’ decisions.

Although the manuscript is thoughtfully written and contributes significantly to university hospitals attracting medical applicants for PGY training, I believe it could benefit from addressing the points outlined below to further enhance its clarity and impact. I hope my comments will be helpful.

Author Response: We appreciate your critical comments and useful suggestions that have helped us improve our paper considerably. As indicated in the responses that follow, we have considered all these comments and suggestions in the revised version of our paper (text highlighted in yellow).

Title:

Comment 2: While the title fits well with the manuscript, I suggest using consistent terminology for "postgraduate training" throughout the document.

Author Response:

Thank you for pointing out the inconsistency in terminology. In response to your suggestion and to ensure clarity, we have standardized the terminology throughout the manuscript. Specifically, we have changes various expressions such as “residency training,” “postgraduate training,” and “postgraduate education” to “postgraduate clinical training.” This term was selected because it aligns not only with your recommendation for consistent usage, but also with the official terminology used by the MHLW to describe the national 2-year training system for medical graduates.

A few representative changes are as follows:

“postgraduate training” → “postgraduate clinical training”

“residency training” → “postgraduate clinical training”

“residency programs” → “postgraduate clinical training programs”

These revisions have been applied consistently throughout the Abstract, Introduction, Results, and Discussion sections.

Abstract:

Comment 3: L.35. I think “Multivariate linear regression” is a misuse term in the abstract. Due to you have only one outcome variable (the number of medical students applying to their alma mater for postgraduate training), “multivariable linear regression” or “multiple regression” is more suitable.

Author Response:

Thank you for pointing out the need for accurate terminology regarding the use of “multivariate.” To better reflect the nature of our statistical analysis—where a single dependent variable was modeled with multiple independent variables—we have replaced all instances of “multivariate” with “multiple regression.” This revision ensures clarity and consistency throughout the manuscript.

The following changes were made:

Page 2, lines 36: “multivariate linear regression analyses” to “multiple linear regression analyses”

Page 2, line 39: “the multivariate analysis” to “multiple regression analysis”

Page 10, lines 210: “a multivariate regression model” to “the multiple regression model”

Table 2. Title: “multivariate” to “multiple regression analyses”

Table 2, Column label: “Multivariate” to “Multiple regression”

Table 2. Note: “The multivariate analysis included...” to “The multiple regression analysis included...”

Page 13, lines 254: “results of multivariate analysis” to “results of multiple regression analysis”

Comment 4: L.31-33. In the abstract, you mentioned that this study would use the variable, the number of medical students applying to their alma mater for postgraduate training. However, in the Data analysis section, you mention that you used “the likelihood of medical…. (L.171-172)” Please make it clear.

Author Response:

Thank you for pointing this out. We agree that the terms “likelihood” and “number” were inconsistent with the actual variable used in the analysis. To avoid confusion and maintain consistency in terminology, we have replaced “likelihood” and “number” with “proportion” in the relevant sections of the manuscript. These modifications were made throughout the manuscript, including in the Abstract, Methods, and Results sections, to ensure consistent use of the term “proportion.”

Introduction:

Comment 5: L.74-75. “the 2022 data from the JRMP……63.5% of trainees 75 opting for community hospitals, compared with only 36.5% choosing university settings.” Is this a special phenomenon being Japan or do other countries also mention this issue? For example, in the United States and Australia, which you had mentioned in the Introduction (L.60).

Author Response:

Thank you for this insightful question. We have clarified that this trend is specific to Japan. Since the 2004 reform of the postgraduate training system, many Japanese medical graduates have preferred community hospitals over university hospitals for residency training owing to factors such as better salaries, work-life balance, and greater hands-on opportunities (Koike et al., 2010; Enari & Hashimoto, 2013). Community hospitals are often perceived as offering stronger clinical exposure, particularly in common diseases, which is supported by higher GM-ITE scores among their residents (Nishizaki et al., 2021). By contrast, in countries such as the United States and Australia, university-affiliated hospitals remain the primary sites for residency training and are valued for their academic structure and prestige (Heist & Torok, 2019; ACGME, 2024). We are not aware of a similar national trend in

---

## [Editor Report · Decision Letter 1]

Factors Influencing Medical Students' Decisions to Remain at Their Alma Mater for Postgraduate Clinical Training: A Cross-sectional

PONE-D-25-12291R1

Dear Dr. Nishizaki,

We’re pleased to inform you that your manuscript has been judged scientifically suitable for publication and will be formally accepted for publication once it meets all outstanding technical requirements.

Kind regards,

Ramya Iyadurai

Academic Editor

PLOS ONE

Additional Editor Comments (optional):

I have no further suggestions or comments
---

## [Editor Report · Acceptance letter]

PONE-D-25-12291R1

PLOS ONE

Dear Dr. Nishizaki,

I'm pleased to inform you that your manuscript has been deemed suitable for publication in PLOS ONE. Congratulations! Your manuscript is now being handed over to our production team.

Kind regards,

on behalf of

Dr. Ramya Iyadurai

Academic Editor

PLOS ONE